# Diagnosis of an Acute Anterior Wall Infarction in Dual-Energy CT

**DOI:** 10.3390/diagnostics13040761

**Published:** 2023-02-17

**Authors:** Aynur Gökduman, Ibrahim Yel, Thomas J. Vogl, Mirela Dimitrova, Leon D. Grünewald, Vitali Koch, Leona S. Alizadeh, Andreas S. Brendlin, Ahmed E. Othman, Simon S. Martin, Tommaso D’Angelo, Alfredo Blandino, Silvio Mazziotti, Christian Booz

**Affiliations:** 1Department of Diagnostic and Interventional Radiology, University Hospital Frankfurt, 60590 Frankfurt, Germany; 2Division of Experimental Imaging, Department of Diagnostic and Interventional Radiology, University Hospital Frankfurt, 60590 Frankfurt, Germany; 3Department of Diagnostic and Interventional Radiology, University Hospital Tübingen, 72076 Tübingen, Germany; 4Department of Diagnostic and Interventional Radiology, University Hospital Mainz, 55131 Mainz, Germany; 5Department of Diagnostic and Interventional Radiology, University Hospital of Messina, 98158 Messina, Italy

**Keywords:** dual-energy CT, myocardial infarct, acute chest pain, cardiovascular diseases, triple-rule-out, virtual monoenergetic images, iodine maps

## Abstract

Due to its high morbidity and mortality, myocardial infarction is the leading cause of death worldwide. Against this background, rapid diagnosis is of immense importance. Especially in case of an atypical course, the correct diagnosis may be delayed and thus lead to increased mortality rates. In this report, we present a complex case of acute coronary syndrome. A triple-rule-out CT examination was performed in dual-energy CT (DECT) mode. While pulmonary artery embolism and aortic dissection could be ruled out with conventional CT series, the presence of anterior wall infarction was only detectable on DECT reconstructions. Subsequently, adequate and rapid therapy was then initiated leading to survival of the patient.

Figure 1Conventional arterial paratransversal CT series (**A**) and reconstructions (**B**,**C**) show multiple soft and calcified plaques in the LAD (RIVA) in the context of CAD. However, the presence of contrast material in the distal course of the LAD remains questionable and cannot be assessed with high diagnostic confidence by conventional CT imaging due to significant image noise and consecutive low image quality.
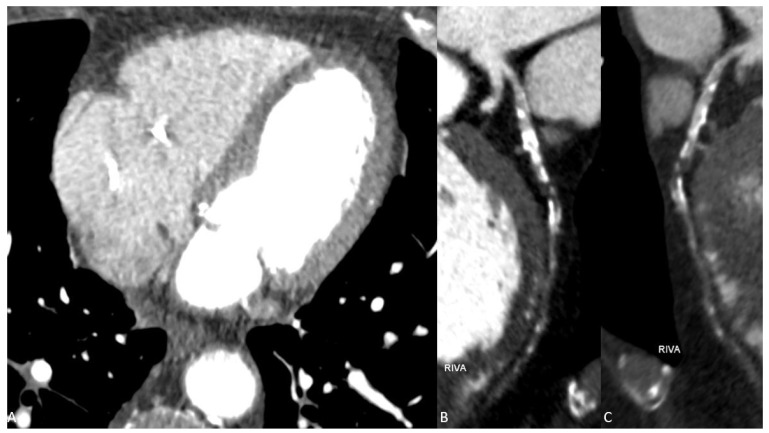

We report the case of a 80-year-old man who presented to our emergency department two days after fall and persistent unclear consciousness. The patient’s medical history included a T-cell lymphoma and a high-grade atrioventricular block treated with a heart pacemaker. Except for chest pain, the patient did not report any medical complaints. Clinically, only a small hemorrhage on the face and ear was noted. Therefore, a cranial CT scan was performed to rule out intracranial hemorrhage. The CT scan showed no evidence of intracranial hemorrhage. Instead, significant reduction in brain volume and a fracture of the nasal bone were detected. After ruling out intracranial hemorrhage, further investigations were performed. The laboratory testing showed elevated levels of the following parameters: C-reactive protein, CK, CK-MB, LDH, Troponin T and D-Dimer. The ECG examination presented acute ST-segment elevation in V1 and V2, suggestive of an anterior wall infarction. The patient was hemodynamically stable and presented improvement in respiration after receiving an anti-infective therapy with Rocephin. Beyond that, a significant decrease in inflammatory parameters was observed. Due to chest pain with syncope and suggestive findings in ECG and laboratory tests with positive D-Dimers and myocardial enzymes, an acute coronary syndrome with syncope was suspected. However, neither an acute pulmonary artery embolism nor an aortic dissection could completely be excluded as differential diagnoses in this constellation. Thus, a triple-rule-out (TRO) DECT scan of the chest was performed. The ECG-gated CT examination was performed on a third-generation, dual-source computed tomography (Somatom^®^ Force, Siemens Healthineers, Forchheim, Germany) consisting of a triple-rule-out scan protocol. According to patient weight, 100 mL of 350 mg iodine/mL (Imeron) was injected into a peripheral vein at a rate of 4 mL/s through a double-piston power injection. After an unenhanced scanning, which is an essential part of our standard protocol, an early contrast-enhanced phase was acquired using test bolus scanning in the descending aorta. The late enhancement scan was obtained 7 min after injection of the contrast medium for myocardial assessment. The total radiation dose was 547 mGycm. CT-scan parameters were as follows: tube A had a voltage of 100 kV and tube current of 165 reference mAs and tube B had a voltage of 140 kV and a tube current of 140 reference mAs. Care Dose4D was applied. All contrast-enhanced CT image series were performed in DECT mode and reconstructed using a medium-soft reconstruction kernel (Qr40f) with a 0.5 mm section collimation in a 0.5 mm increment in the axial, coronal and sagittal planes. In addition, a weighted mix of DECT reconstructions from the 100 kV and 140 kV image data (with a tin filter) were generated to simulate conventional 120 kV images (simulated 120 kV) in all planes. DECT image series were post-processed on a DECT workstation (syngo.via, version VB10B, Siemens Healthineers) to generate virtual monoenergetic images at 55 keV and iodine maps using default settings. While the conventional CT series demonstrated multiple soft and calcified plaques in the left anterior descending (LAD) artery with questionable presence of contrast media in the distal course (Figure 1), the VMI series at 55 keV clearly showed a total LAD occlusion (Figure 2).

Furthermore, the DECT-based iodine maps demonstrated significant hypoperfusion of the anterior wall and anterior septum suggestive for an acute anterior wall infarction, which was missed on conventional CT series (Figure 3). Pulmonary artery embolism and aortic dissection were excluded.

To initiate significant treatment of the patient, percutaneous transluminal coronary angioplasty (PTCA) was subsequently planned and mechanical recanalization with implantation of double stents in the LAD and LCX was performed. Postinterventional echocardiography revealed an ejection fraction of 35%, indicating continuing moderately reduced systolic left ventricular function. In addition to dual antiplatelet therapy and heart failure therapy, an upgrade to an implantable cardioverter defibrillator (ICD) was initiated. The nasal bone fracture caused by the syncope was treated conservatively during the inpatient stay. In order to detect long-term complications, the patient was scheduled for a second coronary angiography four months later and was subsequently transferred to successful geriatric rehabilitation. Follow-up coronary angiography revealed restenosis of the previously deployed stent in the midportion of the LAD. A stent-in-stent deployment was initiated to redilate the stenosis. As a result, left ventricular function also improved. No further complications occurred in the patient. 

Globally, myocardial infarction is the leading cause of death due to its high morbidity and mortality. According to recent data [1], there is an incidence of 290 myocardial infarctions per 100,000 inhibitors, while the mortality rate is about 60 out of 100,000 inhibitants. In order to increase the chances of a person’s survival, it is of enormous importance for to begin treatment as soon as possible after an appropriate diagnosis [2]. Occlusion of the coronary arteries can result in cardiac arrest and functional disorders and lead to death if no adequate treatment is initiated. An anterior wall infarction can affect a large area of the heart [3]. The symptoms of anterior wall infarction are not always pathognomonic and cannot be distinguished from the symptoms of pulmonary artery embolism and aortic dissection or gastritis in certain cases. In addition, other heart diseases such as myocarditis can be the cause of the symptoms. In this context, myocardial anterior wall infarction not infrequently causes diagnostic difficulties but requires rapid workup because of its high lethality [4]. The conventional diagnosis of acute myocardial infarction is usually made with the help of laboratory tests, ECG, and, if necessary, echocardiography. If these examinations do not provide sufficient information about the present disease and do not allow the exclusion of other diseases, a non-invasive CT scan of the heart can be conducted in stable patients if no invasive coronary angiography is performed. In addition, cardiac MRI may be considered as cross-sectional imaging technique for myocardial tissue assessment and functional/volumetric imaging in stable situations, particularly in follow-up [5]. However, this imaging modality is not performed in the acute setting. In the present case, the incident occurred at least 24 to 48 h ago. The performed laboratory tests and the clinical presentation were not strictly specific for a myocardial infarction. Pulmonary embolism and aortic dissection could not be excluded with certainty based on the laboratory results and clinical features. Because of the patient’s hemodynamically stable condition, a DECT scan was preferred over an initial invasive coronary angiography. The DECT examination is our department´s key standard protocol for eliminating differential diagnoses if a final diagnosis of myocardial infarction, pulmonary embolism and aortic dissection has not been made. Using the TRO method, DECT has the advantage that possible differential diagnoses can be successfully excluded. In the context of the presented DECT application, simultaneous visualization of coronary arteries, pulmonary arteries and thoracic aorta in arterial and venous contrast phases, as well as reconstruction of iodine maps and monoenergetic images, allowed the exclusion of pulmonary artery embolism and aortic dissection. Compared with CT angiography, the use of venous-phase DECT as an accurate imaging method for the diagnosis of pulmonary artery embolism has already been significantly demonstrated and represents an additional benefit of DECT in acute coronary pain [6]. While the diagnosis of LAD occlusion and acute anterior wall infarction could not be made with conventional CT images, DECT-based VMI series and iodine maps enabled the prompt detection and therapy initiation of an acute wall infarction due to LAD occlusion. Compared with single-energy CT, DECT has been shown to allow for better tissue differentiation and assessment through the application of a second X-ray spectrum and dedicated postprocessing algorithms [7,8]. In this context, VMI series at low keV levels, such as 55 keV, allow for an increased iodine signal, particularly in vessels, while iodine maps show the direct iodine uptake in tissue without unenhanced background density [9,10]. DECT has made enormous progress in the imaging of cardiovascular diseases in recent years and promises a comprehensive, noninvasive examination of patients in the acute setting [11]. By using CT imaging, nontraumatic as well as traumatic diseases of the chest can be detected. In this context, the combined usage of DECT postprocessing algorithms led to a prompt diagnosis and therapy initiation in our case. However, only very few studies have evaluated the DECT technique and its algorithms in acute myocardial infarction, as well as the significance of iodine mapping for the detection of myocardial perfusion defects to date. In our patient, myocardial infarction and infarct location were reliably diagnosed by DECT, whereas laboratory examination and ECG were nonspecific. In addition, DECT facilitated a prompt diagnosis with high diagnostic confidence compared to conventional CT in our case. Therefore, our recommendation is to perform CT scans in dual-energy mode if technically possible to obtain a rapid and correct diagnosis in comparable circumstances.

## Figures and Tables

**Figure 2 diagnostics-13-00761-f002:**
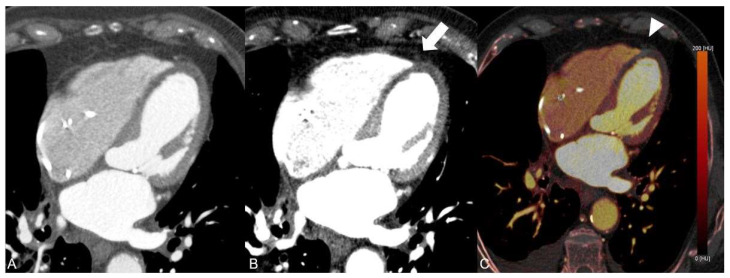
In comparison with the conventional arterial CT image (**A**), the complete lack of iodine in the distal course of the LAD can be visualized by application of dual-energy CT-based virtual monoenergetic images at 55 keV (**B**, arrow) and corresponding iodine maps (**C**, arrowhead).

**Figure 3 diagnostics-13-00761-f003:**
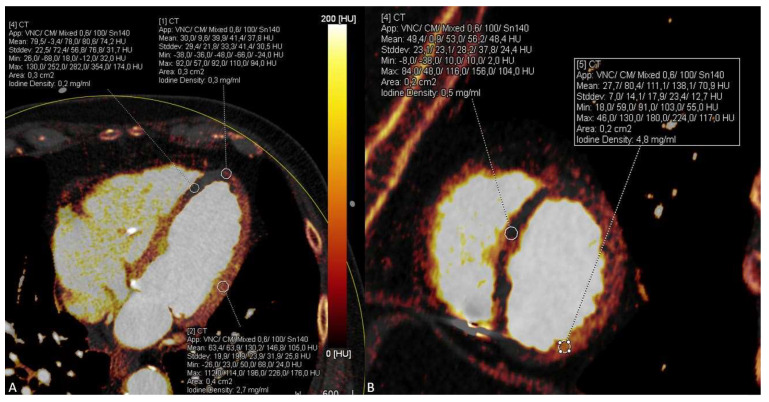
Significant hypoperfusion of the anterior wall and anterior septum is visible and can be quantified on dual-energy CT iodine maps using the four chamber view (**A**) and short axis view (**B**) indicating acute myocardial infarction in this area due to the LAD occlusion.

## Data Availability

In this article, all relevant data are presented. Further inquiries can be directed to the corresponding author.

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
