# Peer review of "Diagnosis of an Acute Anterior Wall Infarction in Dual-Energy CT"

_diagnostics, 2023, doi:10.3390/diagnostics13040761_

Round 1
Reviewer 1 Report
More cases are given if possible
Author Response
Comment 1: More cases are given if possible
Response: Thank you for this feedback. In the period since the presented work, no further cases have been recorded. This is a case reportmanus focusing on one specific situation. We hope you agree with our decision. We would like to cordially thank you for your suggestion to improve our manuscript.
Reviewer 2 Report
ABSTRACT
Ok
KEY WORDS
Ok
CASE REPORT
Well described. DECT protocol is clearly described.
Unenhanced scan was acquired here; I suggest to discuss this choice later in discussion section.
AT page 5 line 100, I suggest to create a new section, named discussion. It seems that clinical data were already described before.
At line 121: I do not agree completely with the sentence: “the performed laboratory tests and the clinical presentation were not specific for a myocardial infarction”. I guess that there was high suspect of infarction, but imaging confirmation was of fundamental importance for subsequent management and that this confirmation was achieved by using a triple rule out exam.
At line 122: I suggest to underline that DECT has the advantage of triple rule out, including the diagnosis of dissection and pulmonary embolism. DECT application of iodine map and MEI+ images have been successfully used for the diagnosis od PE, not only using standard pulmonary artery imaging, but also using aortic or even venous phase imaging of the chest [Identification of pulmonary embolism: diagnostic accuracy of venous-phase dual-energy CT in comparison to pulmonary arteries CT angiography
G Foti. European Radiology 31, 1923-1931]. This topic could be better discussed here. Among possible implications, the need of scanning always in DECT mode, especially when dealing with enhanced scan.
Finally CT can be used to rule out other chest traumatic and non-traumatic conditions (pneumothorax for example) associated with chest pain.
Please add a comment regarding the choice of baseline scan.
Author Response
Comment 1: Unenhanced scan was acquired here; I suggest to discuss this choice later in discussion section.
Response: Thank you for this feedback. An unenhanced scan is an essential component of cardiac CT examinations, especially to detect calcifications of the coronary vessels and to determine the calcium score. In addition, it is useful to first perform a native CT to identify structural and anatomic changes before applying the contrast agent. Therefore, our standard protocol spaced an unenhanced CT scan. According to your suggestion, we have incuded this information in the line 49.
Comment 2: AT page 5 line 100, I suggest to create a new section, named discussion. It seems that clinical data were already described before.
Response: There is a lot of merit to the idea of an outline in subsections. We thank you very much for the suggestion.The subtitles have been removed by the editors and the article type has been formatted from case report to interesting images. Therefore, the subtitles are not desired.
Comment 3: At line 121: I do not agree completely with the sentence: “the performed laboratory tests and the clinical presentation were not specific for a myocardial infarction”. I guess that there was high suspect of infarction, but imaging confirmation was of fundamental importance for subsequent management and that this confirmation was achieved by using a triple rule out exam.
Response: This expression means that the laboratory results are not only specific for myocardial infarction. A high suspicion of infarction was present, but it was not possible to exclude the differential diagnoses of pulmonary artery embolism and aortic dissection on the basis of the standard investigations and to initiate specific therapy. To obtain a definite diagnosis, we performed a DECT examination in the hemodynamically stable patient, which is not normally part of the standard protocol of the confirmed diagnosis of myocardial infarction. We have gratefully included your suggestion on line 122 of the revised version of the manuscript.
Comment 4: At line 122: I suggest to underline that DECT has the advantage of triple rule out, including the diagnosis of dissection and pulmonary embolism. DECT application of iodine map and MEI+ images have been successfully used for the diagnosis of PE, not only using standard pulmonary artery imaging, but also using aortic or even venous phase imaging of the chest [Identification of pulmonary embolism: diagnostic accuracy of venous-phase dual-energy CT in comparison to pulmonary arteries CT angiography
Response: Thank you for this comment. There has been approval of the suggestion in line 128.
Comment 5: G Foti. European Radiology 31, 1923-1931]. This topic could be better discussed here. Among possible implications, the need of scanning always in DECT mode, especially when dealing with enhanced scan.
Response: Your study proposal is greatly appreciated. In our case report, we have incorporated this issue of discussion (line 136).
Comment 6: Finally CT can be used to rule out other chest traumatic and non-traumatic conditions (pneumothorax for example) associated with chest pain.
Response: Thank you for this comment. In line 147 the suggestion has been approved.
Comment 7: Please add a comment regarding the choice of baseline scan.
Response: We have added this aspect to the revised version of the manuscript in line 126.
We would like to thank you for pointing out areas for improvement in our manuscript as well as for the overall positive feedback.